# Layer-Wise High-Impact Parameter Ratio Optimization in Post-Training Quantization for Large Language Models

## Abstract

Large language models (LLMs) have significantly advanced natural language processing, but their massive parameter counts create substantial computational and memory challenges during deployment. Post-training quantization (PTQ) has emerged as a promising approach to mitigate these challenges with minimal overhead. While existing PTQ methods can effectively quantize LLMs, they experience substantial accuracy loss at extremely low bit-widths, primarily due to high-impact parameters that significantly influence quantization performance. Several approaches address these issues by identifying and retaining the high-impact parameters in FP16 format. However, they apply fixed ratios of high-impact parameters across all layers, overlooking layer-wise sensitivity variations. In this paper, we propose a quadratic optimization framework that determines layer-specific ratios of high-impact parameters while considering inter-layer dependencies. We quantize high-impact parameters to moderate bit-widths, which often result in negligible performance degradation in quantized LLMs, while the remaining parameters can be quantized to extremely low bit-widths. Under the same resource-constrained budget, this allows for preserving more high-impact parameters than methods that keep selecting a few in FP16 format. Additionally, the proposed framework allows us to leverage an advanced quantization method that often requires extensive learnable parameters solely for high-impact parameters, while applying a computationally efficient method to the rest. Our approach achieves an effective balance between computational efficiency and model accuracy while maintaining high performance compared to state-of-the-art methods.

## 1 Introduction

Large language models (LLMs) (Wei et al., 2022a; Touvron et al., 2023; Zhang et al., 2022), have gained significant attention due to their remarkable performance in handling complex natural language tasks (Hendrycks et al., 2020), such as language generation, translation, question answering, and text summarization. However, the significantly large size of these models, often comprising billion-level parameters, demands significant computational resources for inference and deployment. To address this issue, an attractive approach is network quantization (Frantar et al., 2023), which not only reduces the computation costs, but also significantly reduces the memory usage. Compared to quantization aware training (QAT) that requires huge training costs and access to large training data, the post-training quantization (PTQ) (Frantar et al., 2023; Lin et al., 2023b; Xiao et al., 2023; Wei et al., 2023), which requires limited calibration data and computational resources, is more in demand for quantizing LLMs.

Previous PTQ methods (Wei et al., 2023; Xiao et al., 2023; Shao et al., 2024) for LLMs have demonstrated the ability to quantize LLMs to lower precision. For example, quantizing weights to 4 bits often results in negligible performance degradation. However, when quantized to extremely low bit-width (e.g., 2-bit weight-only quantization), these methods will lead to significant performance degradation compared to full-precision models. Several approaches (Lin et al., 2023b; Kim et al., 2023; Shao et al., 2024; Cui & Wang, 2024) indicate that high-impact parameters that significantly influence quantization performance are often the biggest challenge in LLM quantization. In CherryQ (Cui & Wang, 2024), the authors separate high-impact parameters in an element-wise manner.

In SqueezeLLM (Kim et al., 2023), the authors represent sensitive weight values in an efficient sparse format and adopt codebook-based non-uniform quantization, which is not hardware-friendly compared to standard uniform quantization methods. Although these approaches successfully improve performance of PTQ for LLMs, applying a fixed ratio of high-impact parameters across all layers overlooks layer-wise variation in parameter importance. Consequently, this approach may be suboptimal as the ratio of importance of parameters differs across layers. Additionally, handling important parameters in an element-wise manner in CherryQ (Cui & Wang, 2024) or using non-uniform quantization as in SqueezeLLM (Kim et al., 2023) may not be hardware-friendly and pose challenges for hardware deployment.

In this paper, we propose a principled approach inspired by (Chen et al., 2021; Deng et al., 2023) for mixed-precision quantization of convolutional neural networks via quadratic optimization to identify the accurate ratio of high-impact parameters for each layer in a channel-wise manner. Specifically, we propose an efficient quadratic optimization approach to determine the optimal ratio of high-impact parameters, which takes into account the impact of different layer sensitivities on model performance. Under the same resource-constrained budget, retaining all high-impact parameters in FP16 format or full precision is not always the best strategy, as it limits the total number of such parameters that can be preserved due to high precision cost. Instead, quantizing high-impact parameters to moderate bit-widths (e.g., 4-bit), the precision that results in negligible performance degradation in quantized models, allows preserving a greater ratio of high-impact parameters for each layer within the budget. Meanwhile, the remaining normal parameters can be quantized to extremely low bit-widths (e.g., 2-bit). Additionally, the more advanced and accurate quantization methods such as AdaRound (Nagel et al., 2020) often lead to better overall results. However, the number of learnable parameters when applying AdaRound is equal to the number of weight parameters, which makes it infeasible to apply AdaRound directly to large language models. To address this, CBQ (Ding et al., 2025) proposes LoRA-Rounding, which considerably reduces the number of learnable parameters. Although this technique achieves substantial improvements for quantizing LLMs, it depends on rank selection, and the significantly reduced parameter space may constrain the search for optimal rounding solutions. Therefore, different from previous methods (Cui & Wang, 2024; Kim et al., 2023; Ding et al., 2025), the proposed approach allows us to adopt a hybrid quantization strategy to balance accuracy and computational efficiency by leveraging advanced quantization methods such as AdaRound only for high-impact parameters, while quantizing the vast majority of remaining parameters using more efficient quantization approaches such as OmniQuant (Shao et al., 2024). This hybrid strategy effectively allocates computational resources based on optimized high-impact parameter allocation, significantly reducing overhead and maintaining overall model quality.

To summarize, the main contributions of this paper are as follows: We propose a novel approach to determine the optimal ratios of high-impact parameters across layers in large language models. Rather than applying a fixed ratio, our approach captures layer-wise sensitivity variations through a quadratic optimization framework. This enables more accurate allocation of quantization precision based on each layer's contribution to overall model performance. Additionally, the proposed approach allows us to utilize advanced PTQ methods that require optimizing a large learnable matrix, such as AdaRound (Nagel et al., 2020), for only the most important parameters. By combining an advanced PTQ method that optimizes a learnable weight-rounding matrix for the most impactful parameters with a lightweight and efficient quantization method for the remaining parameters, our approach achieves a strong balance between performance and efficiency through optimized high-impact parameter allocation.

## 2 RELATED WORKS

**Quantization for large language models.** Quantization is an effective approach to compress LLMs, which helps to significantly reduce inference cost and memory usage. Many studies (Liu et al., 2023b; Shao et al., 2024; Wei et al., 2022c; 2023) show the presence of significantly high-impact parameters in LLMs, and these parameters make the quantization process more challenging and require special handling. One common approach is to use an equivalent transformation to handle the high-impact parameters in LLMs. Other approaches (Xiao et al., 2023; Shao et al., 2024; Lin et al., 2023b) propose shifting transformations and scaling transformations to further improve the performance of quantized models. Recently, rotation transformations (Tseng et al., 2024; Ashkboos et al., 2024; Liu et al., 2025) have been proposed to handle high-impact parameters in LLMs. Several

works also explore mixed-precision quantization (Kim et al., 2023; Lee et al., 2024; Cui & Wang, 2024) to achieve a better trade-off between accuracy and efficiency.

**Handling high-impact parameters for PTQ in LLMs using mixed-precision quantization.** As previously mentioned, high-impact parameters are widely found in the activations and weights of large language models, posing a significant challenge for quantization. Consequently, many mixed-precision methods for LLMs aim to represent a small number of high-impact parameters in higher precision while quantizing other values in lower precision. In AWQ (Lin et al., 2023b), the authors indicate that not all weights in an LLM are equally important, and protecting only a small percentage of high-impact weights can greatly reduce quantization error. LLM.int8() (Dettmers et al., 2022) focuses on quantizing parameters with a mixed-precision decomposition scheme, representing high-impact parameters in 16-bit precision and other values in 8-bit precision. In CherryQ (Cui & Wang, 2024), the authors highlight the importance of cherry parameters, which have a significant impact on quantization performance. They measure parameter impact and identify the most important of parameters for each layer in element-wise manner. They then keep these important parameters in FP16 and then fine-tune the model using quantization-aware training. In SqueezeLLM (Kim et al., 2023), the authors use non-uniform quantization to assign higher bit-widths to more important parameters. While they achieve substantial performance improvements in extreme low-bitwidth quantization, applying a uniform ratio across all layers does not take into account the variation in parameter importance throughout the network, potentially leading to suboptimal allocation of precision resources. In contrast, our method addresses high-impact parameters by taking into account the layer-wise dependencies of the models.

**Advanced quantization for quantizing large language models.** In standard PTQ, rounding-to-nearest is the most common approach as it minimizes quantization error in weight space. However, recent state-of-the-art methods (Nagel et al., 2020; Li et al., 2021; Liu et al., 2023a; Wei et al., 2022b; Jeon et al., 2023) demonstrate that optimizing the rounding function with respect to the task loss significantly improves quantization performance. AdaRound (Nagel et al., 2020) introduces a differentiable rounding mechanism that adapts to data distribution and task objectives, showing remarkable performance improvements for convolutional neural networks (Wei et al., 2022b; Lin et al., 2023a; Jeon et al., 2023), diffusion models (Shang et al., 2023; Li et al., 2023; Huang et al., 2024), and LLMs (Ding et al., 2025). However, applying AdaRound directly to LLMs presents significant challenges, as the number of learnable parameters equals the number of weights in the model, making optimization computationally prohibitive at scale. To overcome this challenge, CBQ (Ding et al., 2025) introduces LoRA-Rounding, which considerably reduces the quantity of parameters that need to be learned. Although this technique achieves substantial improvements for quantizing LLMs, it depends on rank selection, and the significantly reduced parameter space may constrain the search for optimal rounding solutions. In this work, the proposed approach to determine the optimal ratio of high-impact parameters for each layer allows us to take advantage of AdaRound's quantization capabilities while maintaining computational efficiency when quantizing LLMs.

## 3 PROPOSED METHOD

### 3.1 PRELIMINARY ANALYSIS

To examine the sensitivity distribution across hidden dimensions (channels) in large language models (LLMs), we compute the average Fisher information of the parameters associated with each hidden dimension and use it as an importance metric. This metric captures how much each dimension contributes to model performance under quantization. Our experiments employ the LLaMA-2-7B model with 128 calibration samples from WikiText-2. As shown in Figure A.1, the importance scores are significantly different across hidden dimensions, showing that influential parameters are concentrated in a few specific dimensions. This observation shows that we can adopt channel-wise strategy for mitigating problems in handling high-impact parameters, which offers greater hardware efficiency than the element-wise methods used in prior work (Cui & Wang, 2024). Moreover, we observe substantial layer-to-layer variability in the distribution of high-impact parameters—certain layers contain more influential parameters than others, which indicates that applying fixed ratios of high-impact parameters across all layers may not be optimal.

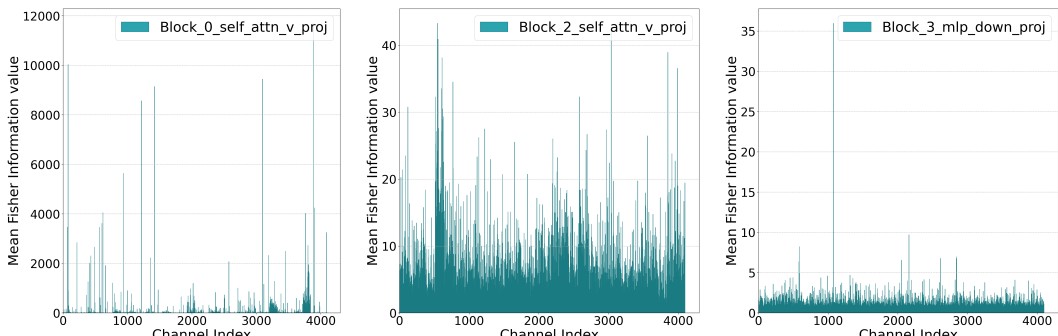

Figure 1: Distribution of average Fisher information across channel dimensions at several layers in the LLaMA-2-7B model. There is a significant difference in importance scores for each layer, indicating that impactful parameters are concentrated in a subset of channels. Additionally, the ratio of high-impact parameters differs across layers.

## 3.2 DETERMINING OPTIMAL RATIO OF HIGH-IMPACT PARAMETERS

Let $\theta_{FP}$ denote the model's full-precision weights and $\theta_Q$ denote the quantized weights. Our goal is to identify the most high-impact parameters in the model to quantize at higher precision, while quantizing the remaining parameters at lower precision. The quantization process for a network's weights can be expressed as $\theta_Q = \text{Quant}(\theta_{FP}) = \theta_{FP} + \Delta$, where $\text{Quant}(\cdot)$ denotes the quantization function and $\Delta$ is the quantization error.

Given the full-precision weights $\theta_{FP}$, the training set $X^{(T)} = \{x_i\}_{i=1}^N$ and the model weight quantization error $\Delta \in \mathbb{R}^{|\theta_{FP}| \times 1}$, the change in the loss of the quantized model can be approximated using a second-order Taylor expansion around $\theta_{FP}$ as follows:

$$\mathcal{L}(\theta_{FP} + \Delta, X^{(T)}) - \mathcal{L}(\theta_{FP}, X^{(T)}) = g^\top \Delta + \frac{1}{2}\Delta^\top \mathbf{H} \Delta + \mathcal{O}(\Delta^\top \Delta), \qquad (1)$$

where $g = \frac{1}{N} \sum_{i=1}^N \nabla_{\theta_{FP}} [\mathcal{L}(\theta_{FP}, x_i)]$ denotes the expected gradient of the loss with respect to $\theta_{FP}$, and $\mathbf{H} = \frac{1}{N} \sum_{i=1}^N \nabla^2_{\theta_{FP}} [\mathcal{L}(\theta_{FP}, x_i)]$ denotes the Hessian matrix. For a well-trained model, $\|g\| \approx 0$ , we can simplify the Eq. (1) to:

$$\mathcal{L}(\theta_{FP} + \Delta, X^{(T)}) - \mathcal{L}(\theta_{FP}, X^{(T)}) \approx \frac{1}{2}\Delta^\top \mathbf{H} \Delta. \qquad (2)$$

In the context of quantization, we define $\mathcal{L}(\theta_Q, X^{(T)}) = \mathcal{L}(\theta_{FP} + \Delta, X^{(T)})$ as the reconstruction loss between the output of the quantized model and that of the full-precision model as follows:

$$\mathcal{L}(\theta_Q, X^{(T)}) = \frac{1}{N} \sum_{i=1}^N \left\| f_{\theta_{FP}}(x_i) - f_{\theta_Q}(x_i) \right\|_F^2, \qquad (3)$$

where $f$ is the output of the model of interest.

**Measure the impact of weight parameters on model performance.** In post-training quantization, a principled approach to assessing parameters' importance is to measure the sensitivity of the training loss to perturbations in individual weight values. This sensitivity is formally captured by the diagonal entries of the Hessian matrix of the loss function with respect to the model parameters. For a given parameter $\theta_{FP,i}$, the corresponding entry $\mathbf{H}_{i,i} = \mathbb{E}_{x_j \sim X^{(T)}} \left[ \nabla^2_{\theta_{FP}} [\mathcal{L}(\theta_{FP}, x_i)] \right]_{i,i}$ represents the expected second-order derivative of the loss, and thus characterizes the local curvature of the loss landscape around the full-precision solution. A larger value of $\mathbf{H}_{i,i}$ implies that the loss is more sensitive to small deviations in $\theta_{FP,i}$, indicating a higher risk of performance degradation under quantization. Therefore, parameters with higher curvature should be preserved with greater precision. We refer to $\mathbf{H}_{i,i}$ as the impact score of the parameter $\theta_{FP,i}$, as it reflects the parameter's relative importance in maintaining model fidelity after quantization.

We approximate the Hessian matrix $\mathbf{H}$ as follows:

$$\mathbf{H}_{i,i} \approx \mathbf{F}_{i,i} = \mathbb{E} \left[ gg^\top \right]_{i,i}. \qquad (4)$$

**Optimal ratio for high-impact parameters.** Following the estimation of the parameter impact, we rank all weight parameters according to their scores. Based on this ranking, we designate a subset of parameters with the highest impact as high-impact parameters, which are more susceptible to quantization error. To preserve model performance, these high-impact parameters are quantized using a higher precision bit-width, denoted as $b^H$, while the remaining parameters are quantized using a standard lower bit-width $b^N$. The proportion of parameters assigned to $b^H$ is treated as an optimization objective and is selected based on the optimization of Eq. (2). Given that the model has $L$ layers and there are $|\mathbb{B}|$ possible options for the high-impact parameter ratio in each layer (where $\mathbb{B}$ is the set of candidate ratios), we have $|\mathbb{B}|^L$ potential configurations of the high-impact parameter ratios across all layers.

For each layer $l$, we define a one-hot vector $\delta_l \in \{0,1\}^{|\mathbb{B}|}$, where $\delta_{l,m} = 1$ indicates that the $m$-th high-impact parameter ratio option is selected. Let $\Delta_{l,m} \in \mathbb{R}^{|\theta_{FP}|}$ denote the difference between the quantized model weight $\theta_Q$ and the full-precision model weight $\theta_{FP}$ when the top $m$-th percentile of weights in layer $l$ are quantized at a higher bit-width, and the remaining weights in the same layer are quantized at a lower bit-width, $|.|$ signifies the cardinality of a given set. The ratio decision vector for the entire network is defined as $\delta \in \{0,1\}^{|\mathbb{B}|L}$, where $\delta = \text{concatenate}(\{\delta_l\}_{l=1}^L)$. The corresponding model weight changes induced by the hybrid quantization scheme are captured by:

$$\Delta = \mathbf{D}^\top \delta, \tag{5}$$

where $\mathbf{D} = \text{concatenate}(\{\Delta_{l,m}^\top\}_{l,m}) \in \mathbb{R}^{L|\mathbb{B}| \times |\theta_{\text{FP}}|}$ is a matrix in which each row $\mathbf{D}_{l|\mathbb{B}|+m}$ represents the weight changes in the $l$-th layer under the $m$-th high-impact parameter ratio option. By combining Eqs. (2) and (5), we can transform the objective in Eq. (2) to:

$$\tfrac{1}{2}\Delta^\top \mathbf{H}\Delta = \tfrac{1}{2}\delta^\top \mathbf{D}\mathbf{H}\mathbf{D}^\top\delta = \tfrac{1}{2}\delta^\top \mathbf{M}\delta, \tag{6}$$

where the matrix $\mathbf{M} = \mathbf{D}\mathbf{H}\mathbf{D}^\top$. In order to identify the optimal layer-wise ratio, we need to approximate the matrix $\mathbf{M}$. By replacing $\Delta$ in Eq. (2) with $\Delta_{l,m}$, we can approximate each diagonal element of $\mathbf{M}$ as:

$$\mathbf{M}_{|\mathbb{B}|l+m, |\mathbb{B}|l+m} \approx \mathbf{D}_{l|\mathbb{B}|+m}\mathbf{H}\mathbf{D}_{l|\mathbb{B}|+m}^\top = \Delta_{l,m}^\top \mathbf{H}\Delta_{l,m} = 2 \cdot (\mathcal{L}(\theta_{FP} + \Delta_{l,m}, X^{(T)}) - \mathcal{L}(\theta_{FP}, X^{(T)})). \tag{7}$$

According to existing works (Li et al., 2021; Ding et al., 2025) that inter-layer interactions are concentrated within individual blocks, we can approximate $\mathbf{H}_{i,j} \approx 0$ if $\theta_{\text{FP},i}$ and $\theta_{\text{FP},j}$ belong to different blocks. Therefore, we have $M_{|\mathbb{B}|l_1+m_1, |\mathbb{B}|l_2+m_2} = \Delta_{l_1,m_1}^\top \mathbf{H}\Delta_{l_2,m_2} = 0$, for any pair of layers $l_1$ and $l_2$ that reside in distinct blocks. For layer pairs within the same block, we can approximate $\mathbf{M}_{|\mathbb{B}|l_1+m_1, |\mathbb{B}|l_2+m_2}$ with:

$$\mathbf{M}_{|\mathbb{B}|l_1+m_1, |\mathbb{B}|l_2+m_2} \approx \mathbf{D}_{l_1|\mathbb{B}|+m_1}\mathbf{H}\mathbf{D}_{l_2|\mathbb{B}|+m_2}^\top = \Delta_{l_1,m_1}^\top \mathbf{H}\Delta_{l_2,m_2}$$
$$= \tfrac{1}{2}((\Delta_{l_1,m_1} + \Delta_{l_2,m_2})^\top \mathbf{H}(\Delta_{l_1,m_1} + \Delta_{l_2,m_2}) - \Delta_{l_1,m_1}^\top \mathbf{H}\Delta_{l_1,m_1} - \Delta_{l_2,m_2}^\top \mathbf{H}\Delta_{l_2,m_2})$$
$$= \mathcal{L}(\theta_{FP} + \Delta_{l_1,m_1} + \Delta_{l_2,m_2}, X^{(T)}) + \mathcal{L}(\theta_{FP}, X^{(T)}) - \mathcal{L}(\theta_{FP} + \Delta_{l_1,m_1}, X^{(T)}) - \mathcal{L}(\theta_{FP} + \Delta_{l_2,m_2}, X^{(T)}). \tag{8}$$

After approximating the elements of matrix $\mathbf{M}$, we optimize the ratio vector $\delta$ by minimizing the quantization error defined in Eq. (2). Let $\mathcal{B}, \mathcal{W} \in \mathbb{R}^{|\mathbb{B}|L}$ be vectors defined by $\mathcal{B}_{i \cdot |\mathbb{B}|+j} = \mathbb{B}_j$ and $\mathcal{W}_{i \cdot |\mathbb{B}|+j} = |\theta_{\text{FP}}^{(j)}|$ for all $0 \le i < L$ and $0 \le j < |\mathbb{B}|$, where $|\theta_{\text{FP}}^{(j)}|$ denotes the number of parameters in the $j^{\text{th}}$ layer of the full-precision model. The overall optimization is defined as:

$$\delta = \underset{\delta}{\arg\min}\ \delta^\top \mathbf{M}\delta \quad \text{s.t.:}\ b^H \delta^\top (\mathcal{B} \odot \mathcal{W}) + b^N \delta^\top ((1 - \mathcal{B}) \odot \mathcal{W}) \le C_{\text{target}},$$
$$\textstyle\sum_{m=1}^{|\mathbb{B}|} \delta_{l,m} = 1, \forall l \in [1, L]\ \wedge\ \delta_l \in \{0,1\}^{|\mathbb{B}|}, \forall l \in [1, L]. \tag{9}$$

Details of the algorithm are provided in Algorithm 1.

### 3.3 HYBRID QUANTIZATION STRATEGY

Based on the optimal ratio for high-impact parameters determined by our quadratic optimization approach, we divide the parameters in each layer into two groups: high-impact parameters and normal parameters with lower impact and apply different quantization strategies to these two groups.

---

**Algorithm 1** High-impact parameters ratio optimization

---

1: **procedure** OPTIMIZE HIGH-IMPACT PARAMETER RATIO($X^{(T)}, \theta_{FP}, L, B, \text{block}(l)$)
2:     ▷ $X^{(T)}$: Calibration data                ◁
3:     ▷ $\theta_{FP}$: Full-precision model          ◁
4:     ▷ L: Number of layers               ◁
5:     ▷ B: Number of high-impact parameter ratio options          ◁
6:     ▷ $block(l)$: Function mapping each layer $l$ to its block index          ◁
7:     initialize the matrix $\mathbf{M}$
8:     **for** $l = 1$ to $L$ **do**          ▷ calculate diagonal elements
9:         **for** $i = 1$ to $B$ **do**
10:            $\mathbf{M}_{|\mathbb{B}|l+i, \, |\mathbb{B}|l+i} \leftarrow 2 \cdot (\mathcal{L}(\theta_{FP} + \Delta_{l,i}, X^{(T)}) - \mathcal{L}(\theta_{FP}, X^{(T)}))$     ▷ Eq. (7)
11:     **for** $l_1 = 1$ to $L$ **do**          ▷ calculate non-diagonal elements
12:         **for** $m_1 = 1$ to $B$ **do**
13:            **for** $l_2 = 1$ to $L$ **do**
14:               **for** $m_2 = 1$ to $B$ **do**
15:                  **if** $block(l_1) \neq block(l_2)$ **then**
16:                     $\mathbf{M}_{|\mathbb{B}|l_1+m_1, \, |\mathbb{B}|l_2+m_2} \leftarrow 0$
17:                  **else**
18:                     $\mathbf{M}_{|\mathbb{B}|l_1+m_1, \, |\mathbb{B}|l_2+m_2} \leftarrow \mathcal{L}(\theta_{FP} + \Delta_{l_1,m_1} + \Delta_{l_2,m_2}, X^{(T)})$
                                      $+ \mathcal{L}(\theta_{FP}, X^{(T)})$
                                      $- \mathcal{L}(\theta_{FP} + \Delta_{l_1,m_1}, X^{(T)})$
                                      $- \mathcal{L}(\theta_{FP} + \Delta_{l_2,m_2}, X^{(T)})$

19:     $\delta \leftarrow \arg\min_\delta \delta^\top \mathbf{M} \delta$          ▷ optimize for the ratio vector in Eq. (9)
20:     **return** $\delta$

---

Given $\theta_Q = \{W_l\}_{l=1}^L$, and for each layer $W_l = \{W_l^H, W_l^N\}$ as the set of high-impact and normal parameters for each layer, respectively. The quantization loss of that layer therefore is defined as:

$$\mathcal{L}(\text{Quant}(W_l), X^{(T)}) = \mathcal{L}(\text{Quant}_H(W_l^H), X^{(T)}) + \mathcal{L}(\text{Quant}_N(W_l^N), X^{(T)}) \qquad (10)$$

where $\text{Quant}_H(.)$ and $\text{Quant}_N(.)$ respectively denotes the quantizer for the high impact and normal hidden dimensions. For the high impact dimensions, we adopt the AdaRound (Nagel et al., 2020) quantization approach that adapts to data and task loss, which is defined as:

$$\text{Quant}_H(W_l^H) = s \times \text{Clip}\left( \lfloor W_l^H / s \rfloor + V_l^H, 0, 2^{b^H} - 1 \right) \quad \text{s.t.: } V_l^H \in [0, 1], \qquad (11)$$

where $s$ is scale, and $V_l^H$ is the learnable weight-rounding parameters with same dimension as $W_l^H$.

During the optimization (Eq. (10)), the elements of the rounding function $V_l^H$ are encouraged toward binary values (0 or 1) using a regularization loss term added to the main objective in Eq. (10):

$$\mathcal{L}_{\text{reg}} = (1 - |2V_l^H - 1|^\gamma), \qquad (12)$$

where $\gamma$ is an annealing factor that starts high and decreases during optimization to encourage convergence to binary values (Nagel et al., 2020).

For the normal parameters, we adopt learnable weight clipping (Shao et al., 2024) as follows:

$$\text{Quant}_N(W_l^N) = \text{Clamp}(\lfloor W_l^N / h \rceil, 0, 2^{b^N} - 1), \text{with } h = {}^{(\alpha \max(W_l^N) - \beta \min(W_l^N))} / {}_{(2^{b^N} - 1)}, \quad (13)$$

where $\alpha$ and $\beta$ are learnable clipping values for the upper and lower bounds of the weight values.

This hybrid approach efficiently allocates computational resources by prioritizing high-impact parameters, greatly reducing overhead while preserving overall model performance.

## 4 EXPERIMENTS

### 4.1 EXPERIMENTAL SETUP

**Calibration dataset and evaluation metrics.** For calibration, we randomly sample 128 sequences, each containing 2048 tokens, from the WikiText-2 dataset (Merity et al., 2016). For evaluation

Table 1: Perplexity scores (↓) of various method evaluated on WikiText-2 and C4 datasets on the settings of *2-bit* quantization on LLaMA-2 models. The results of GPTQ, AWQ and OmniQuant are from (Shao et al., 2024).

| Model | Method | Avg. bit | W2A16 | | Avg. bit | W2A16g128 | | Avg. bit | W2A16g64 | |
|---|---|---|---|---|---|---|---|---|---|---|
| | | | C4 | Wiki2 | | C4 | Wiki2 | | C4 | Wiki2 |
| LLaMA-2-7B | FP16 | 16 | 6.97 | 5.47 | 16 | 6.97 | 5.47 | 16 | 6.97 | 5.47 |
| | GPTQ | 2.00 | – | – | 2.15 | 33.70 | 36.77 | 2.30 | 19.40 | 20.85 |
| | OmniQuant | 2.00 | 90.64 | 37.37 | 2.15 | 15.02 | 11.06 | 2.24 | 12.72 | 9.62 |
| | CBQ | 2.00 | - | – | - | – | – | 2.24 | 11.30 | 8.01 |
| | **Ours** | 2.20 | **14.64** | **9.40** | 2.25 | **11.46** | **8.13** | 2.30 | **10.82** | **7.77** |
| LLaMA-2-13B | FP16 | 16 | 6.47 | 4.88 | 16 | 6.47 | 4.88 | 16 | 6.47 | 4.88 |
| | GPTQ | 2.00 | – | – | 2.15 | 20.97 | 28.14 | 2.30 | 12.48 | 22.44 |
| | OmniQuant | 2.00 | 26.76 | 17.21 | 2.15 | 11.05 | 8.26 | 2.24 | 10.05 | 7.56 |
| | **Ours** | 2.20 | **14.38** | **9.86** | 2.25 | **10.06** | **7.41** | 2.30 | **9.37** | **6.96** |

metrics, we report perplexity (lower is better) specifically on WikiText-2 (Merity et al., 2016) and C4 (Raffel et al., 2020) for language modeling tasks. Following current SOTA works (Shao et al., 2024; Kim et al., 2023), we also evaluate in the zero-shot tasks, including Commonsense reasoning: HellaSwag (Zellers et al., 2019), PIQA (Bisk et al., 2020), WinoGrande (Sakaguchi et al., 2021), and World knowledge: ARC-easy/challenge (Clark et al., 2018). We compare our method against several state-of-the-art PTQ methods: GPTQ (Frantar et al., 2023), AWQ (Lin et al., 2023b), SqueezeLLM (Kim et al., 2023), CBQ (Ding et al., 2025), and OmniQuant (Shao et al., 2024).

**Implementation details.** We evaluate our method on several widely-used LLMs, including LLaMA-2-7B and LLaMA-2-13B (Touvron et al., 2023). Following OmniQuant (Shao et al., 2024), we employ a block reconstruction loss function and validate both per-group and per-channel weight quantization. The notation *W2A16g64* denotes the quantization with 2-bit per-group weight-only, FP16 activation and a group size of 64. All experiments were conducted on NVIDIA A100 GPUs. We set the candidate ratio set $\mathbb{B} = \{0.02, 0.05, 0.1, 0.15, 0.2\}$ when optimizing for the high-impact parameter ratio $\delta$ in Eq. (9). For 2-bit quantization, we use a higher-precision bit-width of 3 bits. Similarly, for 3-bit quantization, we set the higher-precision bit-width to 4 bits. For quantizing the high-impact parameters, we optimize the learnable rounding parameter for 5,000 iterations. The optimizer used is Adam with a learning rate of $10^{-3}$ and an L2 regularization of $10^{-5}$. The annealing factor $\gamma$ in Eq. (12) is initially set to 20 and gradually decreased to 2 during optimization. For the remaining parameters, we adopt the learnable weight clipping method from OmniQuant (Shao et al., 2024).

## 4.2 EXPERIMENTAL RESULTS

**Evaluation on generation datasets with perplexity.** Results in Tables 1 and 2 show the performance of our method in text generation on C4, WikiText-2 using weight-only quantized LLaMA models. Our method consistently outperforms existing methods like AWQ and OmniQuant (Shao et al., 2024), particularly at the low-bit configurations, such as 2-bit and 3-bit, on the LLaMA-2-7B model. Specifically, in the W2A16 setting (Table 1), OmniQuant only achieves a perplexity of 37.37 while our method substantially drives this result to a perplexity of 9.40 on WikiText-2 dataset. Our method also improves the perplexity in the W2A16g64 setting with a gap of 0.24 and 0.48 compared to CBQ (Ding et al., 2025) on WikiText-2 and C4 dataset, respectively. This indicates that extremely low-bit weight quantization requires a careful adjustment for each parameter. In the W3A16 setting (Table 2), the proposed approach improves OmniQuant by 0.25 and 0.36 perplexity scores on WikiText-2 and C4 datasets, respectively.

**Evaluation on downstream tasks.** Table 3 shows the evaluation of our method in multiple zero-shot benchmarks for LLaMA-2-7B at various quantization settings (i.e., W2A16g128 and W3A16g128). In the W2A16g128 setting, the proposed method achieves an average accuracy of 49.04%, outperforming OmniQuant by 1.45% and GPTQ by 10.51%. In the W3A16g128 quantization setting, our method achieves an average accuracy of 60.76%, significantly surpassing GPTQ that performs at 52.39%

Table 2: Perplexity (↓) of *3-bit* quantization on LLaMA-2 models. The results of OmniQuant and AWQ are from (Shao et al., 2024). The results of SqueezeLLM are from (Kim et al., 2023).

| Model | Method | Avg. bit | W3A16 | | Avg. bit | W3A16g128 | | Avg. bit | W3A16g64 | |
|---|---|---|---|---|---|---|---|---|---|---|
| | | | C4 | Wiki2 | | C4 | Wiki2 | | C4 | Wiki2 |
| LLaMA-2-7B | FP16 | 16 | 6.97 | 5.47 | 16 | 6.97 | 5.47 | 16 | 6.97 | 5.47 |
| | GPTQ | 3.00 | – | – | 3.15 | 8.28 | 6.74 | 3.30 | 8.20 | 6.62 |
| | AWQ | 3.00 | – | – | 3.15 | 7.84 | 6.24 | – | – | – |
| | OmniQuant | 3.00 | 8.65 | 6.58 | 3.15 | 7.75 | 6.03 | 3.24 | 7.63 | 5.97 |
| | SqueezeLLM | 3.13 | – | – | – | – | – | 3.24 | 7.51 | 5.96 |
| | CBQ | 3.00 | – | – | – | – | – | 3.24 | 7.56 | 5.89 |
| | **Ours** | 3.13 | **7.84** | **6.03** | 3.25 | **7.55** | **5.85** | 3.30 | **7.49** | **5.82** |
| LLaMA-2-13B | FP16 | 16 | 6.47 | 4.88 | 16 | 6.47 | 4.88 | 16 | 6.47 | 4.88 |
| | GPTQ | 3.00 | – | – | 3.15 | 7.24 | 5.63 | 3.30 | 7.10 | 5.56 |
| | AWQ | 3.00 | – | – | 3.15 | 6.94 | 5.32 | – | – | – |
| | OmniQuant | 3.00 | 7.44 | 5.58 | 3.15 | 6.98 | 5.28 | 3.24 | 6.91 | 5.24 |
| | SqueezeLLM | 3.13 | – | – | 3.24 | 6.82 | 5.23 | – | – | – |
| | **Ours** | 3.13 | **7.08** | **5.32** | 3.25 | **6.78** | **5.22** | 3.30 | **6.80** | **5.16** |

Table 3: The accuracy of 5 common sense reasoning tasks (↑) on LLaMa-2 models.

| Bitwidths | Methods | PiQA | ArcE | ArcC | HellaSwag | WinoGrande | Avg. |
|---|---|---|---|---|---|---|---|
| FP16 | - | 78.07 | 76.34 | 43.51 | 57.17 | 69.21 | 64.87 |
| W2A16 g128 | GPTQ | 58.21 | 33.75 | 19.79 | 29.60 | 51.30 | 38.53 |
| | OmniQuant | 64.79 | 51.13 | 24.83 | 40.30 | 56.90 | 47.59 |
| | **Ours** | **68.82** | **51.30** | **26.19** | **41.65** | **57.22** | **49.04** |
| W3A16 g128 | GPTQ | 76.65 | 73.69 | 40.52 | 54.43 | 66.61 | 52.39 |
| | OmniQuant | 77.37 | 68.01 | 37.20 | 54.21 | 66.30 | 60.62 |
| | **Ours** | **77.80** | **68.51** | **37.37** | **54.33** | **66.32** | **60.87** |

accuracy. In the W3A16g128 quantization, the proposed method reduces the accuracy gap between the quantized and the full-precision models to approximately 4%, demonstrating the effectiveness of our approach in terms of preserving model performance even under low-bit quantization.

## 4.3 ABLATION STUDIES

**The effectiveness of the determining ratio of high-impact parameters.** In this section, we analyze the impact of our proposed approach for determining the ratio of high-impact parameters on model performance. We conduct experiments on LLaMA-2-7B with W2A16 quantization using two settings: (1) a fixed ratio of high-impact parameters across all layers, and (2) layer-specific ratios determined by our approach. In both settings, we quantize both high-impact parameters and remaining parameters using OmniQuant. As shown in Table 4, the layer-specific ratios determined by our approach improves performance by 2.4% on average across downstream tasks and decreases perplexity by 0.15 compared to the fixed-ratio approach. These results demonstrate the effectiveness of the proposed quadratic optimization framework in determining the optimal ratio of high-impact parameters across layers.

**The effectiveness of the hybrid quantization strategy.** To validate the impact of the hybrid quantization strategy on the quantized model performance, we conduct ablation studies on the LLaMA-2-7B with W2A16 quantization. As shown in Table 4, the hybrid quantization strategy improves the performance of LLMs when used with either fixed or optimized ratios of high-impact parameters. Specifically, when combining the fixed ratio of high-impact parameters with the hybrid

Table 4: Ablation study on LLaMA-2-7B with W2A16 quantization. We report WikiText-2 perplexity and average accuracy across downstream tasks.

| Methods | PiQA | ArcE | ArcC | HellaSwag | WinoGrande | Avg. (↑) | PPL (↓) |
|---|---|---|---|---|---|---|---|
| FP16 | 78.07 | 76.34 | 43.51 | 57.17 | 69.21 | 64.87 | 5.68 |
| OmniQuant | 56.20 | 32.78 | 22.44 | 29.36 | 50.91 | 38.34 | 37.27 |
| + uniform ratio | 64.25 | 48.61 | 26.10 | 38.67 | 52.33 | 45.99 | 10.66 |
| + optimal ratio | 67.08 | 52.27 | 27.56 | 41.03 | 53.42 | 48.27 | 9.80 |
| + uniform ratio + hybrid quantization | 66.38 | 49.24 | 26.51 | 39.73 | 53.38 | 47.05 | 10.2 |
| + optimal ratio + hybrid quantization (**Ours**) | **67.14** | **53.89** | **28.10** | **41.13** | **53.49** | **48.75** | **9.40** |

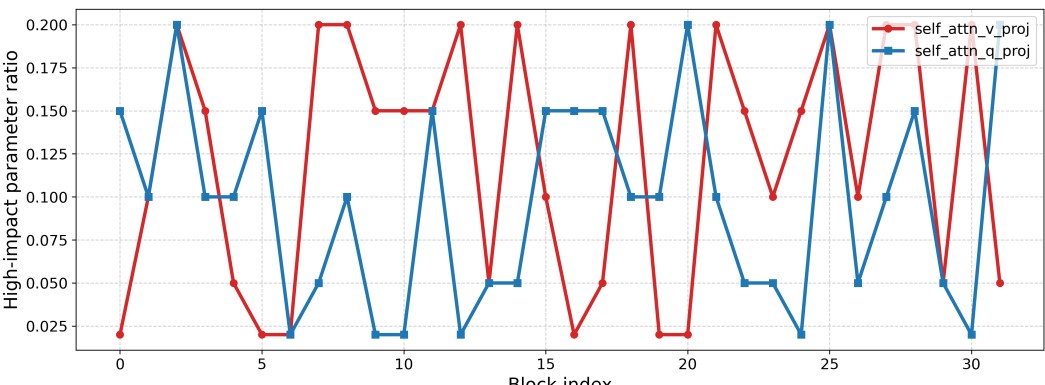

Figure 2: High-impact parameter ratio for selected layers across blocks in the LLaMA-2-7B model with W3A16 quantization.

quantization strategy, the performance of LLMs improves by 1.06% on average across downstream tasks, while enhancing perplexity by 0.46 on average compared to using only the fixed ratio of high-impact parameters. Moreover, the performance of quantized models further improves when combining the optimized ratio of high-impact parameters with the hybrid quantization strategy. These improvements are 2.76% on average across downstream tasks and 1.22 perplexity compared to using only the fixed ratio of high-impact parameters. These results indicate the effectiveness of the hybrid quantization strategy.

**Visualization of optimized high-impact parameter ratio $\delta$ across layers.** Fig. 2 shows the visualization of the distribution of high-impact parameter ratio $\delta$ across layers of the LLaMA-2-7B model in W3A16 quantization setting. In general, the proposed approach allocates various ratios of high-impact parameters to the same layer type across different blocks. Additionally, the values of these ratios are often higher in the *self-attention value (v) projection layer* compared to the *self-attention query (q) projection layer*. This indicates that certain layers contain a larger proportion of high-impact parameters, thus may have a higher influence to the performance of the quantized model.

## 5 CONCLUSION

In this paper, we addressed the post-training quantization for large language models, with a particular focus on the critical issue of high-impact parameters that significantly influence quantization performance. Specifically, we proposed a quadratic optimization framework that determines the optimal layer-specific ratios of these high-impact parameters. This approach enables more accurate allocation of quantization precision based on each layer's contribution to the overall model performance. Furthermore, the proposed approach allows us to apply a hybrid quantization strategy for post-training quantization on LLMs, which leverages advanced quantization methods exclusively for high-impact parameters, while employing computationally efficient methods to remaining parameters. This strategy achieves an effective trade-off between computational efficiency and model accuracy. Extensive experimental results across various model sizes and quantization configurations demonstrate that our approach outperforms state-of-the-art methods, particularly in extremely low bit-width scenarios.

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

**The statement on the use of large language models.** Large Language Models (LLMs) were used solely for grammar correction and language polishing of this manuscript. All research ideas, experimental design, and data analysis were conducted entirely by the authors, and the use of LLMs does not impact the reproducibility or validity of our findings.

# A APPENDIX

## A.1 LIMITATION

While the proposed method improves performance over existing approaches, it relies on manually assigning predetermined high bit-widths to parameters identified as high-impact. This manual configuration may not yield optimal results under a given resource constraint. Future work could focus on dynamically setting the bit-width for these high-impact parameters during training, enabling more efficient and adaptive quantization strategies.

## A.2 ACTUAL INFERENCE SPEEDUP AND MEMORY REDUCTION

## A.3 COMPLEXITY

Regarding time complexity, our framework is designed for efficiency by limiting inter-layer dependency analysis to layers within the same block (see Algorithm 1). For a network with $\mathcal{B}$ blocks, each containing $L_b$ layers, the method performs only $\frac{L_b(L_b-1)}{2}$ validation evaluations per block. As a result, the overall time complexity is $O(\mathcal{B}L_b(L_b - 1))$, which scales linearly with the number of blocks. This block-wise design ensures that the method remains highly efficient and practical, particularly for large-scale networks composed of many small blocks.

## A.4 MORE EXPERIMENTAL RESULTS

Table A.1: Perplexity ($\downarrow$) of *3-bit* quantization on LLaMA-1-7B model.

| Models | Method | Avg. bit | Wiki2 | C4 |
|---|---|---|---|---|
| | – | 16.00 | 5.68 | 7.08 |
| | SqueezeLLM | 3.05 | 6.20 | 7.67 |
| LLaMa-1-7B | SqueezeLLM | 3.24 | 6.13 | 7.56 |
| | SpQR | 3.24 | 6.01 | 7.45 |
| | OmniQuant | 3.15 | 6.15 | 7.75 |
| | Ours | 3.24 | **5.89** | **7.35** |

We conduct experiments using the LLaMA-1-7B model and compare it with other quantization methods. SpQR (Dettmers et al., 2024) and SqueezeLLM (Kim et al., 2023) employ mixed-precision quantization to preserve high-impact weights. Additionally, it introduces non-uniform quantization, allocating more bits to the most high-impact parameters. As shown in Table A.1, we can find that given the same resource constraint, our approach yields better performance than SpQR and SqueezeLLM.

# B MORE ABLATION STUDIES

**Ablation study on the choice of bit-width for high-impact parameters given the same resource constraint.** Given the same resource constraint, we compare the performance of our approach with different bit-widths for high-impact parameters. As shown in Table A.2, given the same resource constraint, when quantizing the model to 2-bit, setting the bit-width for high-impact parameters to 3 achieves better performance than setting it to 4. When quantizing the model to 3-bit, setting the bit-width for high-impact parameters to 4 achieves better performance than setting it to 16.

**Ablation study using different calibration datasets.** We perform experiments using different calibration datasets, including WikiText2 and C4, each consisting of 128 samples with 2048 tokens. As shown in Table A.3, the variance in perplexity scores when quantized with WikiText2 and C4 is small.

Table A.2: The perplexity of quantized LLaMa2-7B models for different high-impact parameter bit-widths.

| Methods | Bit-width | Higher bit-width $b^H$ | Avg. bit | Wikitext2 (PPL ↓) | C4 (PPL ↓) |
|---|---|---|---|---|---|
| OmniQuant | | - | 2.0 | 37.37 | 90.64 |
| | W2A16 | 16 | 2.2 | 15.39 | 26.93 |
| Ours | | 4 | 2.2 | 11.05 | 15.14 |
| | | 3 | 2.2 | **9.40** | **14.64** |
| OmniQuant | | - | 2.0 | 11.06 | 15.02 |
| | W2A16g128 | 16 | 2.25 | 10.05 | 13.51 |
| Ours | | 4 | 2.25 | 8.90 | 11.98 |
| | | 3 | 2.25 | **8.13** | **11.46** |
| OmniQuant | | - | 2.0 | 9.62 | 12.72 |
| | W2A16g64 | 16 | 2.3 | 9.12 | 11.92 |
| Ours | | 4 | 2.3 | 8.25 | 11.01 |
| | | 3 | 2.3 | **7.77** | **10.82** |
| OmniQuant | | - | 3.0 | 6.58 | 8.65 |
| Ours | W3A16 | 16 | 3.13 | 6.26 | 8.16 |
| | | 4 | 3.13 | **6.03** | **7.84** |
| OmniQuant | | - | 3.0 | 6.03 | 7.75 |
| Ours | W3A16g128 | 16 | 3.25 | 6.03 | 7.71 |
| | | 4 | 3.25 | **7.55** | **5.85** |
| OmniQuant | | - | 3.0 | 5.97 | 7.63 |
| Ours | W3A16g64 | 16 | 3.3 | 5.93 | 7.57 |
| | | 4 | 3.3 | **5.82** | **7.49** |

Table A.3: Perplexity scores of LLaMA-2-7B models using different calibration datasets.

| Calibration Dataset | W2A16g128 | | W3A16g128 | |
|---|---|---|---|---|
| | WikiText2 | C4 | WikiText2 | C4 |
| WikiText2 | **8.13** | 11.46 | **7.55** | 5.85 |
| C4 | 8.25 | **11.23** | 7.62 | **5.80** |

## B.1 MORE VISUALIZATION

We visualize the distribution of average Fisher information across channel dimensions for selected layers in the LLaMA-2-7B model. As shown in Figure A.1, the attention query projection layer at block 1 and mlp down projection layer at block 5 exhibit channels with significantly higher importance scores compared to others, while the overall ratio of high-impact parameters could be small. In contrast, the key projection layer in block 2 and the value projection layer in block 5 may contain a higher ratio of high-impact parameters than the previously mentioned layers.

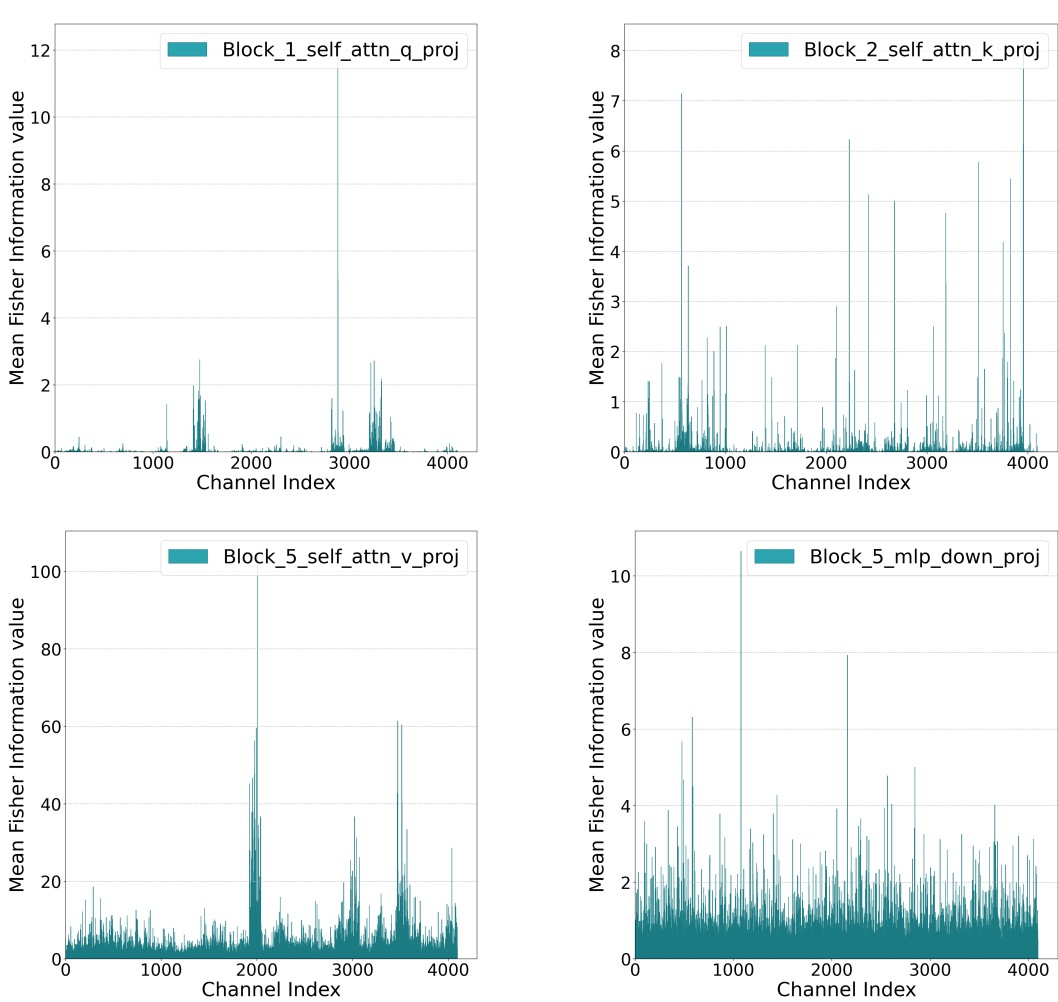

Figure A.1: Distribution of average Fisher information across channel dimensions at several layers in the LLaMA-2-7B model. There is a significant difference in importance scores for each layer, indicating that impactful parameters are concentrated in a subset of channels. Additionally, the ratio of high-impact parameters differs across layers.

