# OpenReview forum: "Layer-Wise High-Impact Parameter Ratio Optimization in Post-Training Quantization for Large Language Models"
_ICLR.cc/2026/Conference — ICLR 2026 Conference Withdrawn Submission_

### Official Review · Reviewer_Nd6P · 2025-10-27

**Soundness:** 3
**Presentation:** 3
**Contribution:** 2
**Rating:** 4
**Confidence:** 4

**Summary:**

The paper introduces a layer-wise quadratic optimization framework to determine the optimal ratio of high-impact parameters in post-training quantization (PTQ) for large language models (LLMs). Unlike prior methods that use a fixed proportion of high-impact parameters across all layers, this approach captures layer-wise sensitivity and allocates bit-widths more precisely. The authors further propose a hybrid quantization strategy, applying advanced methods (e.g., AdaRound) only to high-impact parameters and efficient quantizers (e.g., OmniQuant) to the rest. Experiments on LLaMA-2 (7B and 13B) show improved perplexity under extremely low-bit quantization.

**Strengths:**

Combining AdaRound only for high-impact parameters is an efficient and good idea. It seems reasonable to apply distinct quantization strategies to important parameters and regular ones.

Empirical results demonstrate improved perplexity over OmniQuant and CBQ, particularly under 2–3 bit quantization.

**Weaknesses:**

Although assigning different quantization strategies to important and regular parameters is an interesting direction, the proposed approach seems relatively complex. Moreover, the paper does not discuss whether the AdaRound and OmniQuant strategies might conflict or interfere with each other.

The main experimental results (Tables 1, 2, and 3) are primarily based on the LLaMA models. Whether the proposed method generalizes well to other models remains to be further investigated.

While efficiency is claimed, the paper provides no concrete inference speed or latency measurements.

The algorithm and optimization notations are somewhat dense, and the presentation could be simplified for readability.

**Questions:**

What is the key distinction between the “advanced quantization methods” applied to *important parameters* and the “more efficient quantization approaches” used for *normal parameters*? Specifically, which feature or mechanism makes the quantization method for important parameters genuinely “advanced”?

---

### Official Review · Reviewer_HBuA · 2025-10-31

**Soundness:** 2
**Presentation:** 2
**Contribution:** 2
**Rating:** 2
**Confidence:** 4

**Summary:**

This work proposes a quadratic optimization framework to determine an optimal, layer-specific ratio for critical parameters, considering inter-layer dependencies. This approach is paired with a hybrid quantization strategy: the identified high-impact parameters are quantized to a moderate bit-width (e.g., 3-bit or 4-bit) using the advanced AdaRound method, while the remaining parameters are quantized to an extremely low bit-width (e.g., 2-bit) using a more efficient method. This strategy allows more critical parameters to be preserved within the same resource budget and is shown to significantly outperform state-of-the-art methods on LLaMA-2 models, especially in 2-bit and 3-bit quantization settings.

**Strengths:**

1. The motivation is clear, using channel-wise mix-precision quantization and seting the high-precision propotion through layer-wise sensitivity.
2. The background introduction is comprehensive.

**Weaknesses:**

1. The experiment results are weak.  This paper focus on weight-only quantization, and should compare with more recent state-of-the-art methods, such as EfficientQAT [1], DB-LLM [2], QUIP# [3], ParatoQ [4].
2. The inference efficiency of proposed mix-precision quantization should be measured.
3. It should compare with other mix-precision quantization method, such as SqueezeLLM, in the same average bits.

[1] Efficientqat: Efficient quantization-aware training for large language models, ACL2025
[2] DB-LLM: Accurate Dual-Binarization for Efficient LLMs
[3] QuIP#: Even Better LLM Quantization with Hadamard Incoherence and Lattice Codebooks
[4] ParetoQ: Improving Scaling Laws in Extremely Low-bit LLM Quantization

**Questions:**

Why high-impact dimensions and normal parameters leverage different optimization methods? Is this for efficiency? However, EfficientQAT [1] have demonstrated that simply train all parameters in block-wise manner can achieve the best performance than other complicate designing of trainable parameters.

---

### Official Review · Reviewer_7rD5 · 2025-11-09

**Soundness:** 2
**Presentation:** 1
**Contribution:** 2
**Rating:** 2
**Confidence:** 4

**Summary:**

This paper addresses the significant performance degradation that occurs when applying post-training quantization (PTQ) to large language models (LLMs) at extremely low bit-widths, such as 2-bit. The authors argue that this issue stems from 'high-impact parameters' that significantly influence model performance, and that the conventional approach of allocating a fixed ratio of these parameters across all layers is inefficient.

To solve this problem, the paper proposes two main contributions:
1. It introduces a quadratic optimization framework that determines the optimal ratio of high-impact parameters for each layer. This framework operates under a total bit budget constraint and considers both layer-wise sensitivity and inter-layer dependencies (within the same block).
2. It adopts a hybrid quantization strategy that quantizes high-impact parameters to a moderate bit-width (e.g., 3-bit) instead of retaining them in FP16. This strategy applies a sophisticated method (like AdaRound) to the small subset of high-impact parameters, while applying an efficient method (like OmniQuant) to the remaining majority of normal parameters.

**Strengths:**

- Significance — Valid Problem Formulation: The paper provides strong motivation by empirically demonstrating the non-uniform distribution of parameter sensitivity (using Fisher information in Fig. 1 & A.1). This clearly shows the limitations of a fixed-ratio allocation and establishes a strong need for the proposed layer-specific approach.
- Originality — Principled Optimization Framework: The paper reformulates the layer-wise ratio selection as a quadratic optimization problem (Eq. 5-9), derived from a second-order Taylor approximation of the quantization loss. This is a principled approach that avoids heuristic-based solutions and formally incorporates inter-layer dependencies (via a block-diagonal approximation) while remaining computationally tractable.
- Quality — Hybrid PTQ Design (Targeted Computation): The strategy of applying a sophisticated method (AdaRound) only to the optimized high-impact subset while using an efficient method (OmniQuant LWC) for the rest is a clever design. It differentially allocates computational (calibration) resources according to parameter importance.
- Clarity & Evidence — Ablation Studies: The paper provides clear ablation studies (Table 4) that isolate the contributions of its two main components: (a) the layer-specific optimal ratio search and (b) the hybrid quantization strategy. This effectively validates the independent value of each contribution.

**Weaknesses:**

1. Issues with Experimental Fairness and Validity
The SOTA comparison results are insufficient to clearly prove the method's superiority.
- Reliance on Cited Results: Most SOTA results in Tables 1 and 2 (for GPTQ, OmniQuant, CBQ, etc.) are cited from other papers, not reproduced by the authors in a controlled environment. This makes a fair comparison difficult due to potential differences in calibration datasets, preprocessing, and implementation details.
- Unfair Average Bit (Avg. bit) Comparison: In the 2-bit setting (Table 1), the proposed method consistently uses a higher average bit-rate than the baselines it compares against. For example, for LLaMA-2-7B (W2A16g64), CBQ uses 2.24 bits (PPL 8.01) while "Ours" uses 2.30 bits (PPL 7.77). The performance gain may be partially attributed to this larger bit budget. A comparison under a strictly controlled average bit-rate is needed.
- Marginal Gains at 3-bit: In the 3-bit setting (Table 2, W3A16g64), the PPL improvement over SqueezeLLM (5.96) and CBQ (5.89) by "Ours" (5.82) is less than 0.1, making the significance of the contribution marginal in this setup.

2. Lack of System Performance and Efficiency Verification (Critical Weakness)
The most critical weakness is the complete absence of any system-level performance reporting, which makes it impossible to verify the practical utility of the proposed method.
- Missing Inference Speed: The authors included a section "A.2 Actual Inference Speedup and Memory Reduction" in the appendix, but this section is entirely blank. Despite claiming to be "hardware-friendly," it is unclear what inference speed (e.g., tokens/sec or latency) the resulting hybrid model achieves. A mixed-precision (e.g., 2-bit/3-bit) model with hybrid kernels (AdaRound/OmniQuant) is non-trivial to accelerate efficiently on standard hardware (e.g., GPUs).
- Unreported Calibration Overhead: The quadratic optimization framework (Algorithm 1) appears to have a high calibration cost, as it requires evaluating the loss for all candidate ratios and pairs of layers within a block to populate matrix $M$. The authors only provide a theoretical complexity ($O(BL_b(L_b-1))$) and do not report the actual wall-clock time (e.g., A100 GPU hours) for this setup, nor do they compare it to the calibration cost of baselines like OmniQuant.

3. Limitations in Methodology and Scope
- Strong Approximation Assumptions: The optimization framework approximates the Hessian matrix as block-diagonal, assuming zero interaction between blocks ($H_{i,j}=0$). This simplifies computation but may distort the optimal ratio selection if inter-block dependencies are strong in practice. The validity of this strong assumption is not empirically justified.
- Inaccurate Description of Related Work (AWQ): In the related work section, the authors group AWQ with methods that "protect a small percentage of high-impact weights," implying FP16 retention. This is inaccurate. AWQ's core mechanism is to use activation-aware channel scaling to quantize all weights to a low bit-width in a hardware-friendly manner, without leaving any parameters in FP16. This mischaracterization of a key SOTA method weakens the paper's positioning.
- Limited Experimental Scope: The evaluation is limited to LLaMA-2 7B/13B models and 5 zero-shot benchmarks. Evaluation on more recent or larger models (e.g., Mixtral, LLaMA-3) or on instruction-following/chat benchmarks is not provided.

**Questions:**

1. Quantify Calibration Cost:
The quadratic optimization framework (Algorithm 1) appears to involve a significant calibration cost. Could you please report the actual A100 GPU wall-clock time required to find the optimal layer-wise ratios ($\delta$) for the LLaMA-2-7B model? We are also interested in its relative cost (e.g., 2.5x) compared to the calibration time of OmniQuant or CBQ in your same experimental environment.

2. Report Actual Inference Speed and Hardware Implementation: (Most Important)
The appendix section A.2 "Actual Inference Speedup..." is blank, making it difficult to assess the method's practicality. Given the resulting mixed-precision (2-bit/3-bit) and hybrid-kernel (AdaRound + OmniQuant) model, please report the actual end-to-end inference speed (tokens/sec) and latency. A comparison against a 4-bit GPTQ or 2-bit OmniQuant model (which use uniform kernels) under identical batch size and sequence length is necessary. Furthermore, please clarify if custom CUDA kernels are required to achieve acceleration, and if so, what is the implementation complexity?

3. Provide Strictly Controlled SOTA Comparison:
The SOTA comparisons in Tables 1 & 2 are problematic due to cited results and an inconsistent Average bit (Avg. bit) budget. (e.g., Table 1, LLaMA-2-7B W2A16g64, CBQ @ 2.24-bit vs. Ours @ 2.30-bit). Could you please provide results where you re-run OmniQuant and CBQ in your own experimental setup under a strictly identical average bit budget (±0.01-bit)?

4. Generalization to Newer Architectures:
The experiments are focused on LLaMA-2. We are interested to know if your quadratic optimization framework remains effective on newer architectures such as LLaMA-3, Mistral, Mixtral, or Qwen.

---

### Official Review · Reviewer_ti2G · 2025-11-10

**Soundness:** 3
**Presentation:** 3
**Contribution:** 3
**Rating:** 6
**Confidence:** 3

**Summary:**

This paper tackles extreme low-bit post-training quantization (PTQ) of LLM weights (e.g., 2–3 bits). The authors observe that “high-impact” parameters are unevenly distributed across layers and channels, and that using a fixed high-precision ratio per layer is suboptimal. They therefore (1) estimate Fisher-based importance, (2) formulate a quadratic optimization problem that, under a global bit budget, selects a per-layer ratio of high-impact parameters from a discrete candidate set, and (3) apply a hybrid quantization scheme where high-impact parameters use a stronger PTQ method at moderate bit-width, and the rest use efficient low-bit uniform quantization. Experiments on LLaMA-2-7B/13B with W2A16/W3A16-like settings show substantial gains over GPTQ, AWQ, SqueezeLLM, CBQ, and OmniQuant in both perplexity and zero-shot accuracy, with ablations indicating that both layer-wise ratio optimization and the hybrid scheme contribute.

**Strengths:**

1. Empirically strong. The gains in the W2A16/W2A16g64 configurations are substantial and consistent across LLaMA-2-7B/13B and multiple benchmarks.
2. Reasonable ablations and interpretability. Ablations disentangle the effect of layer-wise ratio optimization from the hybrid quantizer choice. Visualizations of learned ratios provide interpretable evidence that different layers receive different high-precision budgets, which aligns with the sensitivity analysis.
3. Sharp problem focus (extreme low-bit LLM PTQ). The paper targets one of the most brittle regimes in LLM PTQ (2–3 bit weights), where many existing methods break down. The improvements reported in this regime are practically important and not just marginal refinements at 4–8 bits.

**Weaknesses:**

1. Limited detail on the optimization solver and scalability. While the quadratic objective and constraints are described, the paper does not fully specify how the discrete quadratic optimization is solved in practice (exact solver vs heuristic, convergence behavior, any approximations).
2. Limited model and task diversity. Experiments focus on LLaMA-2-7B/13B and standard language modeling + commonsense reasoning benchmarks. There are no experiments on other types of models  (e.g., qwen-series). Similarly, more diverse tasks (e.g., code) could better demonstrate generality.
3. Bit-width choices are still manually specified. The bit-width used for high-impact parameters (e.g., 3 or 4 bits) is manually chosen rather than learned or jointly optimized. In principle, there may be better trade-offs that allocate both ratio and bit-width per layer or per group. This is acknowledged as a limitation but remains unexplored.

**Questions:**

See weakness, further:
1. Comparison to higher-bit OmniQuant variants under matched budgets. Your method mixes extremely low-bit weights with a moderate bit-width for high-impact parameters under a global bit budget. Have you evaluated stronger OmniQuant baselines that are allowed a slightly higher nominal bit-width (e.g., uniform 3-bit, or a 2–3 bit mixed-precision setting) but tuned so that their effective memory/bit budget matches yours?
2. If you train your optimization with different calibration datasets (e.g., different subsets, different domains, or different random seeds), how stable are the resulting per-layer ratios?
3. How does the optimization time and memory overhead scale with the number of layers and the size of the candidate ratio set?
For larger models (e.g., 34B/70B), do you envision using the same formulation, or would you need approximate solvers (e.g., block-wise optimization, low-rank approximations for the interaction matrix, or iterative local search)?

---

### Note · Authors · 2025-11-21

I have read and agree with the venue's withdrawal policy on behalf of myself and my co-authors.